# How Inverse Conditional Flows Can Serve as a Substitute for Distributional Regression

**Lucas Kook**[1]    **Chris Kolb**[2,3]    **Philipp Schiele**[2]    **Daniel Dold**[4]    **Marcel Arpogaus**[4]    **Cornelius Fritz**[5]

**Philipp F. Baumann**[6]    **Philipp Kopper**[2,3]    **Tobias Pielok**[2,3]    **Emilio Dorigatti**[2,3]    **David Rügamer**[2,3]

[1]Institute for Statistics and Mathematics, Vienna University of Economics and Business,
[2]Department of Statistics, LMU Munich, [3]Munich Center for Machine Learning (MCML), [4]HTWG Konstanz,
[5]Department of Statistics, The Pennsylvania State University, [6]KOF Swiss Economic Institute, ETH Zurich

## Abstract

Neural network representations of simple models, such as linear regression, are being studied increasingly to better understand the underlying principles of deep learning algorithms. However, neural representations of distributional regression models, such as the Cox model, have received little attention so far. We close this gap by proposing a framework for distributional regression using inverse flow transformations (DRIFT), which includes neural representations of the aforementioned models. We empirically demonstrate that the neural representations of models in DRIFT can serve as a substitute for their classical statistical counterparts in several applications involving continuous, ordered, time-series, and survival outcomes. We confirm that models in DRIFT empirically match the performance of several statistical methods in terms of estimation of partial effects, prediction, and aleatoric uncertainty quantification. DRIFT covers both interpretable statistical models and flexible neural networks opening up new avenues in both statistical modeling and deep learning.

## 1 INTRODUCTION

Many fundamental statistical modeling approaches, such as random forests or generalized additive models, focus on predicting the (conditional) mean [Wedderburn, 1974, Wood, 2017, Breiman, 2001]. While these approaches comes with extensive theoretical guarantees, they largely ignore aleatoric uncertainty, i.e., the stochasticity in the conditional outcome distribution. Recent developments therefore increasingly model the entire conditional distribution instead of its conditional mean [Kneib et al., 2023]. Motivated by their universal approximation property, neural networks based on Gaussian mixtures were proposed early

to learn conditional outcome distributions [Bishop, 1994]. Approaches to model (all) distributional parameters of a parametric distribution as a function of features have been proposed in statistics and have gained more popularity only in recent years [see Kneib et al., 2023, for details]. An alternative to the aforementioned parametric distributional regression methods was developed in the form of semi-parametric transformation models, which remove the restrictive assumption of a parametric outcome distribution by employing a feature-dependent transformation of the outcome to a simple base distribution [Cheng et al., 1995, McLain and Ghosh, 2013, Hothorn et al., 2014]. This modeling approach is closely related to the concept of normalizing flows in deep learning [Rezende and Mohamed, 2015, Papamakarios et al., 2021].

The idea behind both normalizing flows and transformation models is to learn a feature-dependent transformation between the outcome and a latent variable with a fixed, simple distribution, such as the multivariate standard normal distribution. However, normalizing flows are more expressive than transformation models due to their non-parametric nature and by relying on deep neural networks. Transforming an outcome to a well-behaved latent scale goes back to Box and Cox [1964] and several works have already pointed out the connection between normalizing flows and transformation models [Baumann et al., 2021, Sick et al., 2021, Ausset et al., 2021, Kook et al., 2022b]. However, no formal connection has been established so far.

We close this gap by proposing a class of conditional flows that can be used as a neural network based substitute for various distributional regression approaches (transformation, survival, and mixture models) in statistics.

**Our Contribution** In this work, we propose an assumption-lean modeling framework termed distributional regression using inverse flow transformations (DRIFT). The proposed framework is built around conditional flows, or equivalently, a neural and non-parametric variant of a transformation model replacing its parametric transformation

function (i.e., the inverse conditional flow) with a monotone neural network. Using this model class, we show how numerous statistical models can be understood as a DRIFT. To obtain interpretable model terms, a neural basis function approach is used for processing features. Models in DRIFT can be treated in a unified maximum likelihood framework, covering continuous, discrete binary, ordered as well as censored outcomes. We compare models in DRIFT with state-of-the-art distributional regression models in real-world applications featuring ordinal, (clustered) time-series and survival outcomes and demonstrate that DRIFT is a competitive alternative. We conclude with a benchmark study in which DRIFT is shown to be a well-working, neural network-based framework for distributional regression in terms of predictive performance. DRIFT thus offers one way to interpolate between statistical models and complex neural networks in terms of flexibility and intelligibility.

## 2 RELATED LITERATURE

**Structured Neural Regression Models**  Structured neural regression bridges the gap between the inherent interpretability of statistical models and the predictive power of black-box neural networks as universal approximators [Hornik et al., 1989]. Early attempts to integrate statistical models and neural networks focused on data exploration [Ciampi and Lechevallier, 1995], combining pre-trained statistical models within a network of models, or using the parameters of statistical models as initial network weights [Ciampi and Lechevallier, 1997], with extensions to generalized additive neural networks [Potts, 1999, de Waal and du Toit, 2007].

Advancements in deep learning, including automatic differentiation and the availability of efficient modular software libraries, have led to the recent introduction of semi-structured distributional regression [Rügamer et al., 2023b]. This approach proposes an end-to-end differentiable hybrid network architecture that combines interpretable structured additive predictors, as seen in GAMs, and arbitrary deep learning models within a distributional regression framework. This has led to a series of subsequent developments, particularly critical in domains such as medicine, where it is essential to model interpretable effects of tabular features alongside unstructured data modalities [Rudin, 2019], such as images [Baumann et al., 2021, Dorigatti et al., 2023, Kook et al., 2022b, Kopper et al., 2022, Herzog et al., 2023].

A distinct approach is proposed in neural additive models [Agarwal et al., 2021] and its extensions [e.g., Chang et al., 2023, Radenovic et al., 2022, Yang et al., 2021]. Here, a standard additive GAM predictor for the conditional mean is assumed, with the shape functions for each feature learned in separate subnetworks. This approach overcomes the potential limitations of pre-defined basis functions to model highly complex or jagged functions, albeit at the cost of drastically increased parameter counts and a still-evolving theoretical foundation [Heiss et al., 2019, Zhang and Wang, 2022]. Despite the recent progress in structured neural regression models, they typically impose the restrictive assumption of a known parametric outcome distribution, highlighting the importance of non-parametric normalizing flows.

**Normalizing Flows**  Normalizing flows model a random variable $Y$ with a complex distribution through compositions of invertible and differentiable transformations of a latent random variable $\varepsilon$ that has a known base distribution with no free parameters, such as a standard normal distribution. In our context, these diffeomorphisms are parameterized by deep neural networks [Dinh et al., 2016]. Specifically, normalizing flows express $Y$ as $\phi(\varepsilon)$, where $\phi : \mathbb{R} \to \mathcal{Y}$ is a flow. Usually, $\phi$ follows a pre-specified functional form that enables fast inversion and computation of the determinant of the Jacobian, which is required to obtain the density of $Y$. Examples include coupling [Dinh et al., 2016], planar and radial [Rezende and Mohamed, 2015], autoregressive [Kingma et al., 2016], or residual flows [Chen et al., 2019]. The weights parametrizing $\phi$ can be learned through maximum likelihood training, and several such transformations can be stacked to approximate arbitrarily complex distributions. Normalizing flows have been successfully applied both to conditional and unconditional generative modeling, and distributional regression [Papamakarios et al., 2021, Winkler et al., 2019].

## 3 ASSUMPTIONS IN STATISTICAL MODELING

Statistical modeling requires direct input by the data analyst in the form of assumptions that express relevant domain knowledge, which is unavoidable in situations with scarce or highly complex data such as multi-task learning [Silver et al., 2013], biology [Xu and Jackson, 2019], material science [Childs and Washburn, 2019], physics [Stewart and Ermon, 2017], and more. Common assumptions fall into two distinct categories.

**Distributional Assumptions**  The construction of both mean and distributional regression models usually requires assuming a known, parametric family of distributions of the underlying outcome. Typical examples in statistical modeling encompass the linear model (Gaussian error distribution), generalized linear and additive models assuming an exponential family [Hastie and Tibshirani, 1986, Nelder and Wedderburn, 1972], or structured additive distributional regression models, specifying additive predictors for all distribution parameters of an *a priori* known parametric distribution [Kneib et al., 2023]. Identifying an appropriate parametric distribution for the model generally depends on either solid domain knowledge or exhaustive model com-

parisons. Moreover, most distributional assumptions do not allow for multimodality in the learned distribution. Unmet assumptions can lead to inconsistent, biased, or inefficient estimation of model parameters [see, e.g., Pawitan, 2001, White, 1982]. To robustify results against misspecified distributions, various approaches try to compensate for unexplained variance or samples, e.g., by applying outlier removal or using a more heavy-tailed distribution [see, e.g., Huber, 2011]. While this practice can improve performance, model validation and diagnostics are manual and iterative processes that, in turn, often require domain knowledge [White, 1981].

**Structural Assumptions**  To foster interpretability and limit complexity, statistical models commonly make additional structural assumptions. Two of the most common ones are additivity and linearity of predictors. This means that the conditional mean of a response $Y$ given features $X$, $\mu(X) := \mathbb{E}[Y|X]$ (or any other aspect of the conditional distribution), relates to features by $g(\mu(X)) = X\beta$ with invertible link function $g$ and feature weights $\beta$. One of the most prominent examples following this assumption is the generalized linear model [Nelder and Wedderburn, 1972]. Extensions of (generalized) linear models, such as GAMs [Hastie and Tibshirani, 1986, Wood, 2017], allow to go beyond linear feature effects by using, e.g., a spline basis representation to introduce non-linearity. In this case, domain knowledge is often needed to choose the best-fitting spline basis, the number and position of knots, or the amount of smoothness [see, e.g., Gu, 2013, Schumaker, 2007, Wood, 2017]. Another typically human-based decision for such models is the inclusion of higher-order feature interactions. Limiting the degree and number of interactions allows controlling the number of parameters (and thus scalability) while ensuring a certain level of interpretability.

# 4 INVERSE CONDITIONAL FLOWS FOR DISTRIBUTIONAL REGRESSION

Consider observations $\{(y_i, x_i)\}_{i=1}^n$ of a univariate outcome $Y \in \mathcal{Y} \subseteq \mathbb{R}$ and features $X \in \mathcal{X}$. In this work, we focus on modeling the entire conditional distribution of $Y$ given $X$. We propose a flexible class of models for the conditional distribution of $Y$ given $X$ that interpolates between highly flexible normalizing flows (low domain knowledge) and parametric models (high domain knowledge) for various outcome types. In this class, a model needs two components to be fully specified. First, a parameter-free base distribution with cumulative distribution function (CDF) $F : \mathbb{R} \to [0, 1]$ and continuous, two times differentiable, log-concave density $f$; and second, a conditional flow $\phi : \mathbb{R} \times \mathcal{X} \to \mathcal{Y}$ which maps observations from the parameter-free base distribution to the conditional outcome distribution for all constellations of features. By conditional flow, we refer to a (composition of) function(s), later parameterized by neural networks,

which is monotonically increasing for all possible realizations of the features.

Let $\mathcal{P}$ denote the class of all conditional CDFs with sample space $\mathcal{Y}$ and conditional on features in $\mathcal{X}$. Then for each base CDF $F$, the class of models under investigation can be defined as the set $\Phi_F$ containing all conditional flows $\phi : \mathbb{R} \times \mathcal{X} \to \mathcal{Y}$ such that for all conditional distributions $F_{Y|X} \in \mathcal{P}$, we have that for $\epsilon \sim F$, $\phi(\epsilon, X) \sim F_{Y|X}$.

Domain knowledge can now enter as restrictions on $\Phi_F$. The conditional cumulative distribution function of $Y$ given $X$, denoted by $F_{Y|X}$, can for all $x \in \mathcal{X}$ be written as

$$F_{Y|X=x}(\cdot) = F(\phi^-(\cdot, x)),$$

where $\phi^-(\cdot, x) := \sup\{z \in \mathbb{R} : \phi(z, x) \leq \cdot\}$ denotes the conditional (generalized) *inverse* flow. The inverse flow plays an important role in training models in DRIFT (see Section 4.3). Next, we consider what types of assumptions can be imposed on these models and how those assumptions affect model capacity, i.e., the flexibility of conditional flows contained in $\Phi_F$ and the conditional distributions they can model (see Figure 1 for an example). Then, we discuss explicit parameterizations of and how to train models in DRIFT.

## 4.1 ASSUMPTIONS ON THE BASE DISTRIBUTION

On their own, assumptions on the base distribution do not limit model capacity, because any conditional CDF $F_{Y|X} \in \mathcal{P}$ can be composed as $F \circ F^{-1} \circ F_{Y|X}$. Then the set of functions

$$\{\phi^- = F^{-1} \circ F_{Y|X} \mid F_{Y|X} \in \mathcal{P}\}$$

gives rise to all conditional flows with base CDF $F$. Choosing a particular $F$ only fixes the scale on which to interpret the components of the flow $\phi$.

**Example 1** (Assumptions on the base distribution). Binary classification via logistic regression can be thought of as DRIFT with standard logistic CDF $F$ and inverse conditional flows on the log-odds scale, i.e.,

$$\phi^-(y, x) = \log \frac{F_{Y|X=x}(y)}{1 - F_{Y|X=x}(y)}.$$

However, the conditional cumulative distribution $F_{Y|X}$ can be modeled with other base distributions, such as the standard minimum extreme value, or standard normal distribution. These correspond to inverse conditional flows interpretable on the log-hazard $\text{cloglog}(\pi) = \log(-\log(1-\pi))$, or probit $F_{N(0,1)}^{-1}$ scale [Tutz, 2011].

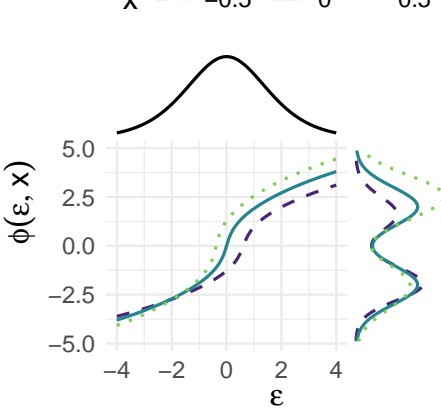

Figure 1: Depiction of the location-scale DRIFT in Example 2. For three values of $X$ (dashed/solid/dotted), the standard logistic base distribution (top side) is transformed into the conditional outcome distribution (right side) via the conditional flows (middle). In this example, the distribution for $x = 0$ is a normal mixture with equal weights (solid line).

## 4.2 STRUCTURAL ASSUMPTIONS

Structural assumptions take a variety of forms and limit model capacity more or less severely. In DRIFT, we impose an additivity assumption on the conditional flow $\phi$ in terms of the features, which is the distribution-free analog to additivity assumptions in GAMs on the scale of the conditional mean. We define the class of location-scale conditional flows used in DRIFT by

$$\Phi_F^{\text{L-S}} = \{\phi \in \Phi_F \mid \phi(\varepsilon, x) = \phi_0(\mu(x) + \sigma(x)\varepsilon)\}, \quad (1)$$

where $\mu : \mathcal{X} \to \mathbb{R}$ controls location, $\sigma : \mathcal{X} \to \mathbb{R}_+$ controls scale, and $\phi_0 : \mathbb{R} \to \mathcal{Y}$ is called *reference flow*, because it is the flow from $F$ to any $F_{Y|X=x_0}$ for which $\mu(x_0) = 0$ and $\sigma(x_0) = 1$ (see Figure 1). Additivity assumptions on $\phi$ restrict model capacity but do not imply a fixed family of conditional outcome distributions because no distribution assumptions are implied by the reference flow. Features can only change location and scale of the base distribution before applying the reference flow [Rezende and Mohamed, 2015, Siegfried et al., 2023].

**Example 2** (Structural assumptions). Here, we illustrate how to construct and sample from a location-scale conditional flow with a single feature. First, we choose a reference flow $\phi_0 := F_{Y|X=x_0}^{-1} \circ F$ for an arbitrary reference $x_0 \in \mathcal{X}$. Then, we introduce a shift $\mu$ and scale effect $\sigma$. Samples from $Y|X = x$ are then generated via $Y := \phi_0(\sigma(x)\epsilon + \mu(x))$. For example, we choose $\epsilon \sim F$ to follow a standard logistic distribution, and a Gaussian mixture $0.5N(-2, 1) + 0.5N(2, 1)$ for $F_{Y|X=x_0}$. Further, we introduce a nonlinear $\mu : x \mapsto \mu(x)$ with $\mu(x) = \exp(1 - \exp(-x)) - 1$ and $\sigma : x \mapsto \sqrt{\exp(x)}$

as the shift and scale effects. Here, $\mu$ and $\sigma$ are such that the reference is $x_0 = 0$ since $\mu(x_0) = 0$ and $\sigma(x_0) = 1$. The interplay between base distribution, conditional flow and conditional outcome distribution in DRIFT is shown in Figure 1.

A combination of distributional and stronger structural assumptions can fix the conditional outcome distribution and thus severely limit model capacity. When limiting the reference flow to the identity, i.e.,

$$\{\phi \in \Phi_F^{\text{L-S}} \mid \phi(\varepsilon, x) = \mu(x) + \sigma(x)\varepsilon\},$$

the distribution of $Y|X$ is limited to the location-scale family induced by $F$. For instance, with $\epsilon \sim N(0, 1)$, $\mathcal{Y} \subseteq \mathbb{R}$ and $\mathcal{X} \subseteq \mathbb{R}^d$, $\mu : x \mapsto (1, x)^\top \beta$ and $\sigma \equiv c > 0$, such linear conditional flows recover linear regression.

## 4.3 TRAINING MODELS IN DRIFT VIA MAXIMUM LIKELIHOOD

Models in the DRIFT framework lend themselves to estimation via maximum likelihood. DRIFT can be used to model the distribution of outcomes with binary, ordinal, count-valued, continuous and mixed discrete-continuous sample space $\mathcal{Y}$. For continuous (exact) responses, the likelihood function is given by the log-density, whereas for discrete responses, the likelihood is obtained as a difference in cumulative distribution functions. Since we have access to the entire conditional distribution, we can also evaluate the likelihood contributions of censored outcomes.

We consider the log-likelihood function for exact continuous, discrete and uninformatively censored outcomes. For exact continuous observations $y \in \mathbb{R}$, the log-likelihood $\ell : \Phi_F \times \mathcal{Y} \times \mathcal{X} \to \mathbb{R}$ is given by the log-density, i.e.,

$$\ell(\phi, y, x) = \log f(\phi^-(y, x)) \frac{d}{dv} \phi^-(v, x)\big|_{v=y}.$$

Using deep learning libraries, involved gradients of the log-likelihood can be computed efficiently. For discrete outcomes supported on $\{y_1, y_2, \ldots, y_K\} \subseteq \mathbb{R}$, we have for $k = 2, \ldots, K - 1$,

$$\ell(\phi, y_k, x) = \log[F(\phi^-(y_k, x)) - F(\phi^-(y_{k-1}, x))].$$

The log-likelihood of interval censored outcomes $(y_l, y_u]$ can be defined likewise,

$$\ell(\phi, (y_l, y_u], x) = \log[F(\phi^-(y_u, x)) - F(\phi^-(y_l, x))],$$

with a slight abuse notation when allowing interval-valued observations.

To evaluate the likelihood, we thus need evaluate the base CDF $F$ and the (generalized) inverse flow $\phi^-$. Since $F$ is fixed, we now turn to parameterizations of $\phi^-$.

## 4.4 PARAMETERIZING MODELS IN DRIFT

We parameterize models in DRIFT explicitly via neural networks. Three components need to be specified: The inverse reference flow $\phi_0^-$, shift $\mu$, and scale $\sigma$ effect. The inverse reference flow needs to be monotonically increasing and its smoothness depends on the outcome type. Location effects are unconstrained, whereas scale effects need to fulfill a simple positivity constraint. In this work, we parameterize all three functions via neural networks.

**Inverse Reference Flow**    For discrete outcome types, a dummy encoded basis with increasing coefficients is sufficient to ensure monotonicity. For absolute continuous outcomes, $\phi_0^-$ can be a smooth invertible function. Classically, $\phi_0^-$ has been parameterized via basis expansions, such as $B$-splines [Hothorn et al., 2014] or polynomials in Bernstein form [Hothorn et al., 2018, McLain and Ghosh, 2013]. Here, we parameterize the reference flow via monotonic neural networks [Huang et al., 2018]. Sufficient conditions for monotonicity are given in the following result.

**Proposition 1** (Monotonicity of the conditional flow).  Consider an inverse conditional flow of the form

$$\phi^-(y, \mathbf{x}) = \phi_{y\mathbf{x}}^-(\phi_y^-(y), \varphi(\mathbf{x})),$$

where $\phi_{y\mathbf{x}}^-$, $\phi_y^-$, and $\varphi$ are feed-forward neural networks. For $\phi^-(y, \mathbf{x})$ to be strictly monotonically increasing in $y$, it is sufficient for $\phi_{y\mathbf{x}}^-$ and $\phi_y^-$ to have strictly positive weights and strictly monotonic activation functions (e.g., tanh activations).

In the special case of parameterization (1), we thus only need to choose $\phi_0^-$ to be a monotonic neural network. The proof of the more general result can be found in, e.g., Silva et al. [2018].

**Location and Scale Effects**    To avoid restrictive structural assumptions while preserving interpretability, we specify predictors $\psi$ for $\mu$ and $\sigma$ using neural basis functions [Agarwal et al., 2021], i.e.,

$$\psi(x) = \sum_{j=1}^{J} \rho_j(x_j), \qquad (2)$$

where each $\rho_j$ represents a feature-specific network learning an adaptive basis function for the respective feature $x_j$. This network can be further extended to, e.g., include bivariate feature effects $\sum_{i,j:i\neq j} \rho_{i,j}(x_i, x_j)$ as also done in our numerical experiments or even higher-order interactions. In case multiple feature effects contain the same feature $x_j$, various approaches exist to ensure the model's identifiability [see, e.g., Rügamer et al., 2023b]. Identifiability is particularly important if the model predictor in (2) is further extended by a more complex (deep) neural network capturing higher-order interaction effects. In that case, the recently proposed approach in Rügamer [2023] provides a non-invasive post-hoc adaption of the model that is also suitable for our approach.

## 5   NUMERICAL EXPERIMENTS

We now present a variety of numerical experiments where we investigate whether DRIFT is a viable substitute to one or more established statistics approaches of similar complexity and aligns with their goodness-of-fit, effect estimation, and predictive performance. These experiments also provide insights into whether normalizing flows can be similarly interpretable as statistical models. In the Supplementary Material, we further analyze the hyperparameter stability of models in DRIFT and give the explicit parameterization of all models and competitors used in the experiments.

**Setup**    In Sections 5.1–5.3, and 5.6, we parameterize $\phi_0^-$ in terms of an invertible neural network. To further demonstrate the ease with which to interpolate between a fully neural and semi-parametric $\phi_0^-$, we specify $\phi_0^-$ in Sections 5.4 and 5.5 using polynomials in Bernstein form, an alternative to monotone neural networks used in transformation models [Hothorn et al., 2018] and also recently discussed for normalizing flows [Ramasinghe et al., 2021].

### 5.1   ORDINAL REGRESSION

The UCI "Wine quality" dataset [Cortez et al., 2009] contains 1599 red wines whose quality is described on an ordinal scale (10 levels of which only 3–8 have been observed). We consider five features, namely fixed and volatile acidity, citric acid and residual sugar content, and concentrations of chlorides. Non-linear effects are specified by a feature-specific ReLU-network with four hidden layers and eight units each.

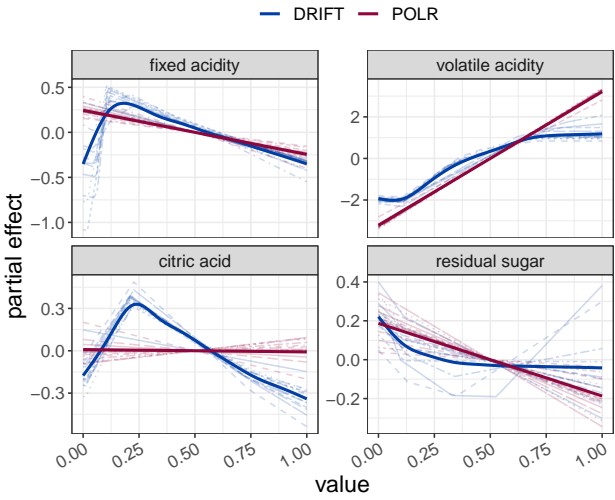

Figure 2: Estimated partial effects for four features in a 20-fold cross-validation of the UCI wine quality dataset using a DRIFT and a proportional odds logistic regression (POLR) model.

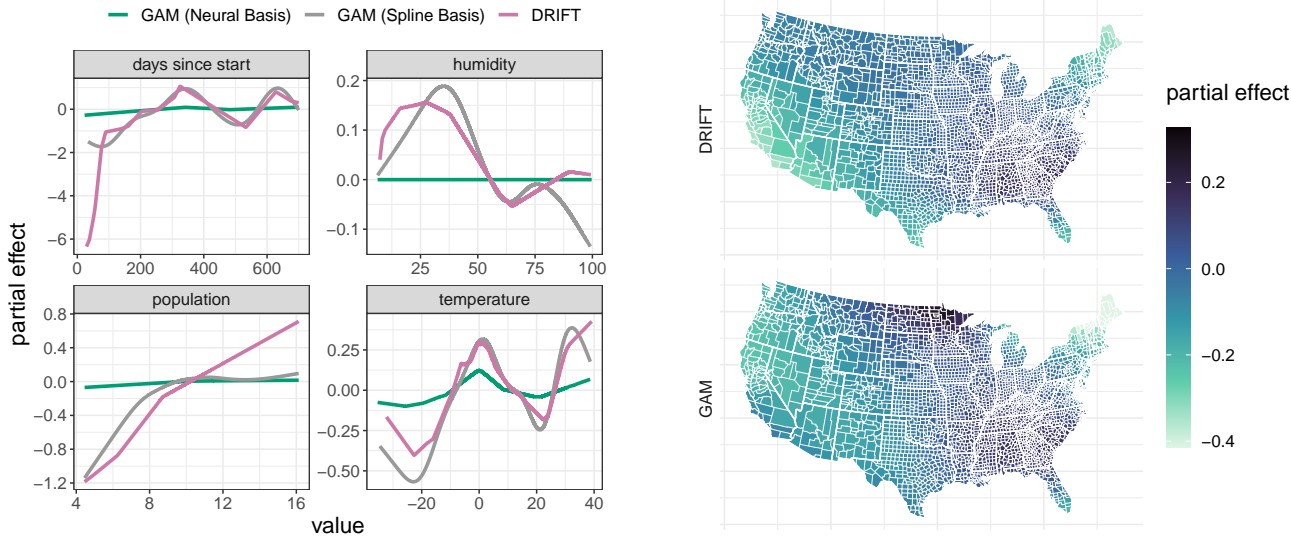

Figure 3: Left: Estimated effects on the prevalence of Covid-19 using a GAM with neural or spline basis and DRIFT (colors). Right: Estimated spatial effects on the prevalence of Covid-19 from DRIFT and GAM (with spline basis).

**Results**: In a 20-fold cross-validation, an ordinal DRIFT performs on par with the standard proportional odds logistic regression with linear feature effects (log-score $-1.11$ (0.06) *vs.* $-1.12$ (0.06)) and partial effect estimates deviate from the proportional odds model (POLR, Figure 2).

### 5.2  GENERALIZED ADDITIVE MODELS

To compare DRIFT with GAMs, we re-analyze a spatio-temporal data set of number of Covid-19 incidences in the US previously analyzed in Rügamer [2023]. We follow their data cleaning procedure and model the prevalence of infections using a GAM (either by using a B-spline basis or feature-specific neural networks) with features for population, date, latitude and longitude, temperature, and humidity. We then compare these previous methods with DRIFT (using a monotone neural network for $\phi_0^-$) qualitatively by analyzing the estimated partial effects.

**Results**: Inspecting Figure 3 (left) we find that the partial effects based on neural basis functions are underfitted compared to spline-based partial effects, and even collapse to a zero effect for humidity. This difficulty in training neural additive models is a well-known phenomenon in the literature. In contrast, effects estimated via DRIFT look very similar to those obtained from a traditional GAM with spline basis. This is also the case for the spatial effect (Figure 3, right).

### 5.3  TIME SERIES ANALYSIS

We forecast the hourly electricity consumption of 370 clients contained in the UCI *Electricity* dataset [Dua and Graff, 2017]. We model each of the univariate time series with

48 consecutive lags based on 9 days of data (starting 2014-07-01). We first determine the optimal number of training epochs using data for the first 7 days as training set and the 8-th day as validation set for early stopping. Then, knowing the optimal number of epochs, we use the first 8 days for the final training and forecast on the 9-th day.

**Results**: We obtain a log-score of -0.538 (0.195). As a comparison, we use the *auto.arima* function [Hyndman et al., 2023] that fits an ARIMA model with an automatic search for the best model parameters. This results in a worse performance with a log-score of -4.434.

### 5.4  MIXTURE MODELING FOR MULTIMODAL DISTRIBUTIONS

We next demonstrate the flexibility of our approach to model multimodal distributions. To this end, we investigate the ATM dataset from Rügamer et al. [2023a], known to follow a time-dependent process with mode-switching behavior. We focus on the multimodality in the data and compare our approach against mixture models – a classical choice for multimodal data. Both approaches use the time information as a feature to allow for a time-varying density estimation.

**Results**: Based on a predefined test dataset, we find that the negative log-likelihood (smaller is better) of our approach is 2.09 whereas the mixture model with 2 components as reported in results in a value of 2.27. When investigating the learned densities (see Figure 4), we find that both approaches capture the 2 modes at later time points, but DRIFT works slightly better by also capturing the multimodality at earlier points in time. Furthermore, our approach does not require to specify the number of modes *a priori*.

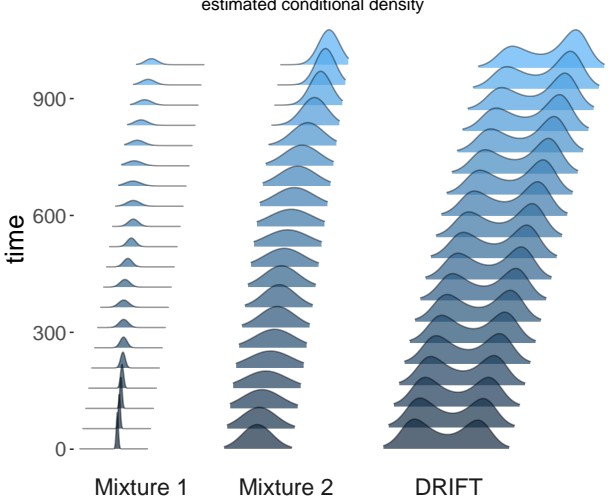

estimated conditional density

Figure 4: Left and center: Estimated Gaussian densities over time (y-axis) for the two mixture components in the mixture model. Right: Estimated densities over time using DRIFT.

## 5.5 SURVIVAL ANALYSIS

Next, we use a DRIFT to learn spatio-temporal determinants of response times (time-to-arrival) of the London fire brigade to fire-related emergency calls. The data has been previously used in Taylor [2017], Kopper et al. [2022]. DRIFT allows various survival analysis model classes to be used, e.g., a piecewise exponential additive model [Bender et al., 2018] or a Cox proportional hazards [Cox, 1972] model. Using 25 different hold-out splits we compare a DRIFT resembling a Cox model with smooth log cumulative baseline hazard $\phi_0^-$ (see Supplementary Material for details) with a piecewise exponential additive model based on the reweighted integrated Brier score [IBS, Sonabend, 2022]. We also review the performance of a featureless learner (Kaplan-Meier) and report the IBS at the quartiles of the follow-up time.

**Results**: Figure 5 shows the comparison between results obtained by a piecewise exponential additive model (PAM) and DRIFT in terms of prediction performance (integrated Brier score; left) and qualitatively in terms of estimated log cumulative hazard functions conditional on time of day. The DRIFT shows slightly better out-of-sample performance in terms of re-weighted integrated Brier score and estimtates a stronger influence of time of day on the survivor curve compared to the piecewise exponential additive model. Martingale residuals [Barlow and Prentice, 1988] on the held-out data for a single split (Figure 5, right) show that PAM and the DRIFT make qualitatively similar prediction errors and illustrate that DRIFT allows residual-based model checks for non-continuous outcomes.

## 5.6 STRUCTURED ADDITIVE DISTRIBUTIONAL REGRESSION AND TRANSFORMATION MODELS: BENCHMARK STUDY

Finally, we check if DRIFT is able to match the predictive performance of other additive distributional regression approaches. For the comparison, we use a structured additive distributional regression with parametric distribution assumption [Klein et al., 2015], a transformation model [Hothorn et al., 2014], and DRIFT for which the reference flow is non-parametric. As both structured additive distributional regression and transformation models represent very flexible approaches that are closely related to our method, we run a benchmark study to compare performances using an extended collection of the classical UCI machine learning repository datasets [Dua and Graff, 2017]. As base distribution $F$ we use a Gaussian for both DRIFT and the transformation model. The distributional regression is also defined based on a Gaussian distribution. The predictors for the transformation and distributional regression model are defined using thin-plate regression splines. For DRIFT we use neural basis function splines based on one fixed architecture (see Supplementary Material for details). For $\phi_0^-$, we use a simple monotonic neural network with two hidden layers of either 10 or 20 neurons each. We also compare these methods when using a semi-structured predictor, i.e., when enhancing the structured predictor with a deep neural network for all features and methods (see details in Appendix B.3.6).

Table 1: Comparison results for different datasets (rows) and the *structured* methods (DR: distributional regression; TM: transformation model, DRIFT: Location-scale) showing the mean log-score (and standard deviation in brackets) based on a 10-fold cross-validation. The best methods per dataset are highlighted in bold.

| Dataset | DR | TM | DRIFT |
|---|---|---|---|
| Airfoil | $-3.6$ (0.3) | $-3.2$ (0.2) | $\mathbf{-3.1}$ (0.1) |
| Concrete | $-3.4$ (0.2) | $-3.5$ (0.3) | $\mathbf{-3.3}$ (0.1) |
| Diabetes | $-5.8$ (0.3) | $-5.6$ (0.4) | $\mathbf{-5.2}$ (0.2) |
| Energy | $-2.7$ (0.2) | $-2.6$ (0.1) | $\mathbf{-2.3}$ (0.1) |
| Fish | $-1.4$ (0.2) | $\mathbf{-1.3}$ (0.3) | $\mathbf{-1.3}$ (0.1) |
| ForestF | $-1.9$ (0.3) | $-1.7$ (0.2) | $\mathbf{-1.4}$ (0.2) |
| Ltfsid | $-6.5$ (0.2) | $-6.2$ (0.3) | $\mathbf{-4.7}$ (0.1) |
| Naval | $4.0$ (0.2) | $3.8$ (0.3) | $\mathbf{5.1}$ (0.1) |
| Real | $-1.3$ (0.4) | $-1.1$ (0.2) | $\mathbf{-0.9}$ (0.1) |
| Wine | $0.5$ (0.2) | $0.8$ (0.4) | $\mathbf{4.2}$ (0.2) |
| Yacht | $-1.5$ (0.3) | $-1.7$ (0.4) | $\mathbf{-0.8}$ (0.1) |

**Results**: Our results (Table 1) confirm that DRIFT is able to match the performance of other methods, in many cases even outperforming them. This is particularly notable for datasets in which the outcome exhibits a non-Gaussian distribution (unconditionally). For example, in the *Wine* dataset, the outcome is technically discrete, but commonly treated as continuous. Here, distributional regression with a para-

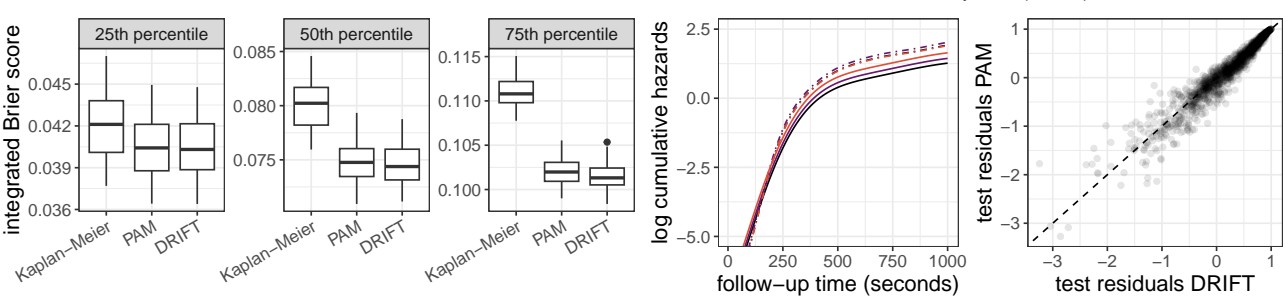

Figure 5: Left: Predictive performance in terms of integrated Brier score (lower is better, evaluated at the 25th, 50th, and 75th percentile) of a Kaplan-Meier estimator, a piece-wise exponential additive model (PAM) and a DRIFT. Middle: Estimated log cumulative hazards given daytime (in hours; colors). Right: Out-of-sample martingale residuals show comparable prediction errors for DRIFT and PAM.

Table 2: Comparison results for different datasets (rows) and *semi-structured* methods (DR: distributional regression; TM: transformation model, DRIFT: Location-scale) showing the mean log-score (standard deviation) based on a 10-fold cross-validation. The best methods per dataset are highlighted in bold.

| Dataset | DR (Semi) | TM (Semi) | DRIFT (Semi) |
|---------|-----------|-----------|--------------|
| Airfoil | **-2.9** (0.1) | -3.0 (0.1) | -3.1 (0.5) |
| Concrete | -3.3 (0.1) | -3.3 (0.3) | **-3.0** (0.3) |
| Diabetes | -5.7 (0.5) | -6.0 (0.4) | **-5.4** (0.2) |
| Energy | -2.9 (0.1) | -2.7 (0.5) | **-2.2** (0.1) |
| Fish | **-1.3** (0.1) | -1.5 (0.2) | **-1.3** (0.2) |
| ForestF | -2.0 (0.3) | -1.9 (0.2) | **-1.4** (0.4) |
| Ltfsid | -7.7 (7.5) | -5.9 (0.7) | **-4.6** (0.1) |
| Naval | 4.1 (1.0) | 3.9 (0.1) | **5.1** (0.3) |
| Real | -1.4 (0.4) | -1.4 (0.6) | **-1.2** (1.1) |
| Wine | -0.2 (0.0) | -0.4 (1.0) | **2.1** (2.1) |
| Yacht | -1.0 (0.2) | -2.2 (2.0) | **-0.5** (0.2) |

metric Gaussian assumption yields the worst results. Using a transformation model can improve this result, however, both parametric alternatives are outperformed by the non-parametric neural reference flow used in DRIFT.

We also run the same comparison as presented in Table 1 when using model formulations discussed in Baumann et al. [2021], Rügamer [2023], namely (i) when combining structured predictors with neural networks (Table 2), (ii) neural basis functions for all three approaches (Table 4 in the Supplement), and (iii) only deep neural network predictors for all three methods (Table 5 in the Supplement). Similar to the results presented in Table 1, DRIFT is on par with or improves upon DR and TM.

## 6 CONCLUSION

We demonstrate that numerous statistical models can be expressed in the DRIFT framework. Equipped with neural ba-

sis functions, DRIFT enables interpretable model terms with little requirement for manual input from the modeler. The versatility and practical applicability of DRIFT is reinforced by favorable benchmark comparisons and applications involving various outcome types. Overall, our results suggest that DRIFT serves as a competitive neural network-based framework for distributional regression tasks. A promising avenue for future research involves developing statistical inference methods tailored to DRIFT.

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

# How Inverse Conditional Flows Can Serve as a Substitute for Distributional Regression (Supplementary Material)

**Lucas Kook**[1]     **Chris Kolb**[2,3]     **Philipp Schiele**[2]     **Daniel Dold**[4]     **Marcel Arpogaus**[4]     **Cornelius Fritz**[5]

**Philipp F. Baumann**[6]     **Philipp Kopper**[2,3]     **Tobias Pielok**[2,3]     **Emilio Dorigatti**[2,3]     **David Rügamer**[2,3]

[1]Institute for Statistics and Mathematics, Vienna University of Economics and Business,
[2]Department of Statistics, LMU Munich, [3]Munich Center for Machine Learning (MCML), [4]HTWG Konstanz,
[5]Department of Statistics, The Pennsylvania State University, [6]KOF Swiss Economic Institute, ETH Zurich

## A  ADDITIONAL EXPERIMENTS

### A.1  MULTIPLE DEFAULTS

To investigate the influence of hyperparameters on the performance of DRIFTs, a grid search is conducted for a large collection of datasets from the UCI repository (details below). The purpose of this study is to find a good default that works well on most datasets such that DRIFTs can be used off-the-shelf similar to other (mostly tuning-free) distributional regression approaches. To account for the stochasticity of the training process, each combination of hyperparameters is trained 3 times with different seeds. Next to different learning rates and dropout rates, the architectures of the neural networks are varied via the number of units and layers for both the features and outcome of the network (i.e., the experiments assume an unstructured predictor for the feature part of the DRIFT). The following hyperparameters are chosen for the grid search:

- learning rate $\in \{10^{-2}, 5 \times 10^{-3}, 10^{-3}\}$,
- dropout $\in \{0, 0.5\}$,
- seed $\in \{1, 2, 3\}$,
- units feature network $\in \{20, 50, 100\}$,
- number of layers feature network $\in \{1, 2\}$,
- units $\phi_0^- \in \{20, 50, 100\}$,
- layers $\phi_0^- \in \{2, 10\}$,
- last layer units $\phi_0^- \in \{5, 20\}$.

The normalized validation negative log-likelihood ($\text{NLL}_{\text{val}}$) for each choice of hyperparameters across data sets resulting from the grid search is displayed as boxplots in Figure 6. It is evident that neither the specific choice of hidden units, the number of layers, nor the learning rate had a consistent effect on the validation loss, with only an increased dropout rate leading to slightly worse results across most datasets. The analysis suggests that the model performance is largely robust with respect to these hyperparameters.

### A.2  INFLUENCE OF INITIALIZATION

While there seems to be little influence in the choice of hyperparameters, architecture and learning rate, we found that the initialization of weights in the monotonic NN $\phi_0^-$ plays a key role. This part of the DRIFT requires special attention as every weight is defined to be positive to guarantee monotonicity. This can, e.g., be implemented in TensorFlow by using the `non_neg` constraint function for every weight in every layer. In addition to the constraint function, the initialization of the weights should also be positive. In the following, we analyze three different initializations. For the first initialization, we set the lower bound of the Xavier initialization Glorot and Bengio [2010] to zero by sampling from $w \sim U\left(0, \sqrt{6/(\text{fan}_{\text{in}} + \text{fan}_{\text{out}})}\right)$, where $\text{fan}_{\text{in}}$ is the number of neurons in the previous layer and $\text{fan}_{\text{out}}$ the number or

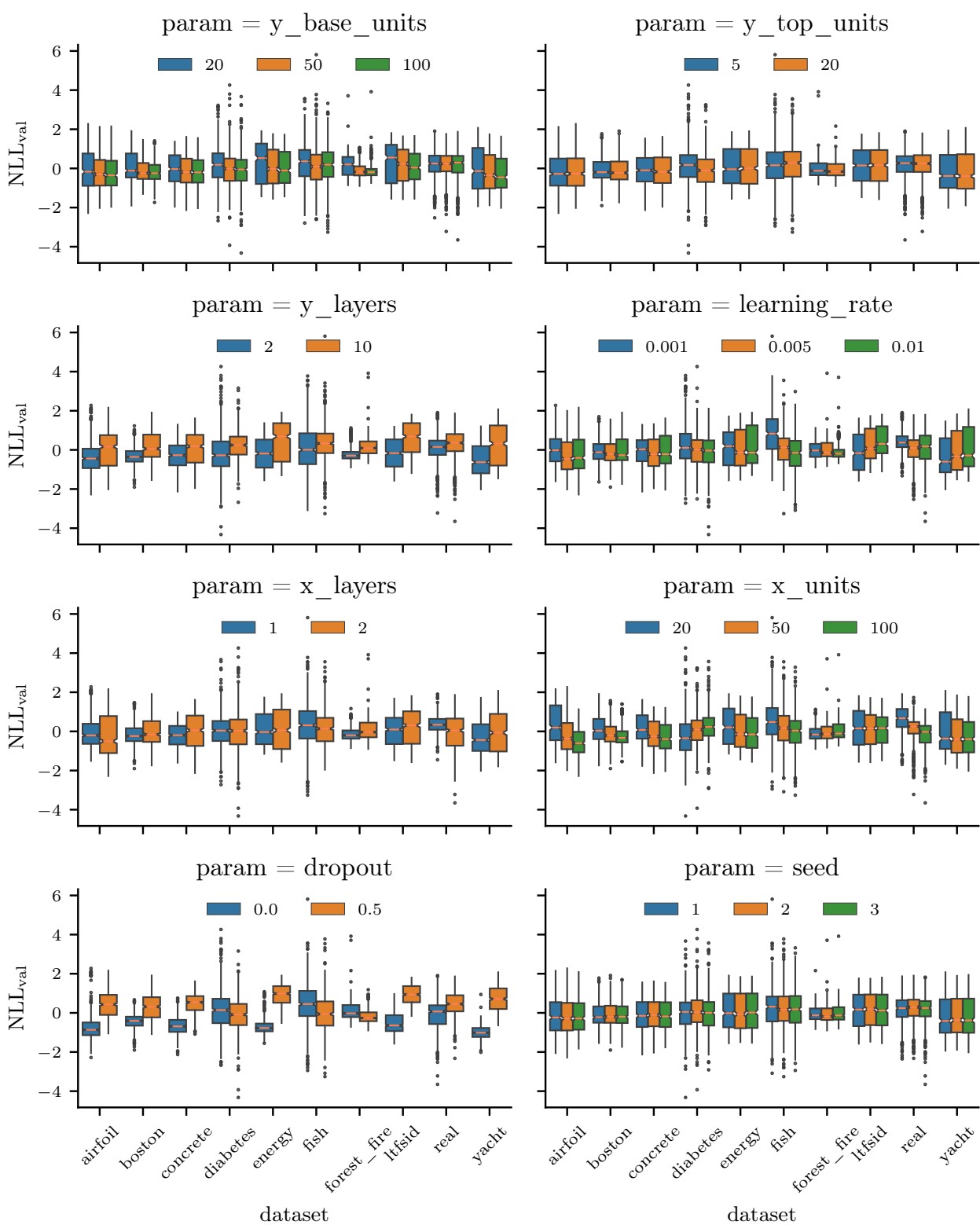

Figure 6: The results of the grid search on different datasets.

neurons in the current layer. Using this naive approach, the variance after each layer increases, and even a rather small neural network (e.g., with three hidden layers) will have difficulty converging. To analyze the effect in more detail, we simulate data

from a standard normal distribution ($Y \sim N(0,1)$) and pass it through the initialized $\phi_0^-$ network. The resulting distribution after each activation function with the Xavier initialization and $\tanh$ activation is shown in the top row of Figure 7. We see that after three hidden layers, the activations in the network have saturated, making the training extremely challenging.

The second initialization is based on the assumption that the expectation and variance after each layer should remain constant. Using this assumption and by using a uniform distribution with a zero lower bound it follows that the upper bound $b$ should be initialized with $b = \sqrt{\frac{3}{\text{fan}_{\text{in}} + \text{fan}_{\text{out}}}}$. The middle row in Figure 7 shows the activation distribution with this initialization and the same input data. We see that this alternative initialization scheme improves the saturation problem to some extent. However, after the third layer, most of the activations are still saturated.

Further adapting the initialization, we empirically find that

$$w \sim U\left(0, \sqrt{\frac{9}{\max(\text{fan}_{\text{in}}, \text{fan}_{\text{out}})^2}}\right) \tag{3}$$

results in only minor changes in the variance between different layers and solves the convergence problems even for deeper architectures.

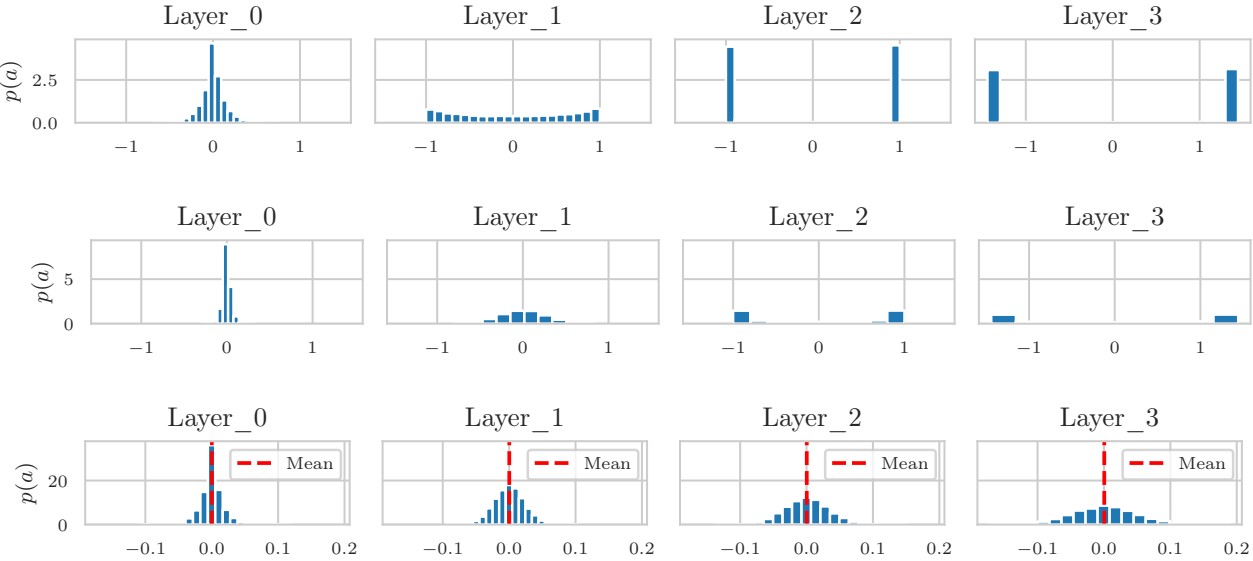

Figure 7: The value distribution of activations after each layer when processing 10,000 samples from a standard normal distribution. The network architecture is based on a four-hidden layer network with 100, 100, 20, and one output neuron, respectively. The top row shows the results by initializing the weights according to $w \sim U\left(0, \sqrt{6/(\text{fan}_{\text{in}} + \text{fan}_{\text{out}})}\right)$, the middle row by using a uniform distribution with upper bound $b = \sqrt{\frac{3}{\text{fan}_{\text{in}} + \text{fan}_{\text{out}}}}$, and the bottom row by using the initializing function according as given in Equation 3.

# B   NUMERICAL EXPERIMENT DETAILS

## B.1   GENERAL PARAMETRIZATION

In most cases where we use DRIFT or neural basis functions, we specify

- $\phi_0^-$ by a monotonic neural network with two hidden layers, each with 10 hidden units, positive weight constraint and a tanh activation function. The hidden layers are followed by an output layer with 1 unit, also with positive weight constraint, but linear activation function to allow mapping into $\mathbb{R}$.

- the neural basis functions $\rho_j$ by a multi-layer perception with a 64-64-31-1 architecture, ReLU activations except for the last layer (which has no activation), and a bias in the penultimate layer only.

## B.2   UCI BENCHMARK DATASETS

Table 3 provides an overview of UCI datasets used in our benchmark.

Table 3: Data set characteristics and references.

| Data set | # Obs. | # Feat. | Pre-processing |
|---|---|---|---|
| Airfoil | 1503 | 5 | - |
| Concrete | 1030 | 8 | - |
| Diabetes | 442 | 10 | - |
| Energy | 768 | 8 | - |
| Fish | 908 | 6 | - |
| ForestF | 517 | 12 | logp1 transformation for `area`; numerical representation for `month` and `day` |
| Ltfsid | 182 | 4 | - |
| Naval | 11934 | 16 | - |
| Real | 414 | 6 | Subtract minimum for `X1`; logp1 transformation for `X2`; log transformation for `X3 − X6` as well as for the outcome |
| Wine | 178 | 13 | - |
| Yacht | 308 | 6 | - |

## B.3   DETAILS OF INDIVIDUAL EXPERIMENTS

### B.3.1   Mixture Modeling

The Gaussian mixture (of experts) model implemented in a neural network as suggested in Rügamer et al. [2023c] is parametrized by a mixture of two normal distributions with the same additive predictor for both mean and standard deviation parameter $\mu_\kappa, \sigma_\kappa, \kappa = 1, 2$. The additive predictor contains an intercept $\beta_{0,\mu,\kappa}$ and $\beta_{0,\sigma,\kappa}$, respectively, and a thin-plate regression spline $f_{\mu,\kappa}, f_{\sigma,\kappa}$, respectively, for the feature $x_{\text{time}}$.

DRIFT is parametrized using a Bernstein polynomial of order 30 for $\phi_0^-$ and a location shift using a neural basis function $\rho(x_{time})$ defined by a feed-forward neural network with two hidden layers, each with 64 units, ReLU activation and a bias term, and a final layer with 1 unit, linear activation and no bias term.

### B.3.2   Ordinal Regression

As in Gal and Ghahramani [2016], we consider the subset of red wines and use the same cross-validation folds. All features were standardized to the unit interval. For each of the 20 splits, we fit a `tram::Polr` Hothorn et al. [2022] with linear covariate effects and a DRIFT via `deeptrafo::PolrNN` Kook et al. [2022a] with a neural basis function architecture.

The neural basis functions are specified via a fully connected neural network with four ReLU layers with eight units each and a single unit last layer with linear activation. The estimated partial effects for each predictor were centered to have mean zero. The DRIFT was trained for 200 epochs with the Adam optimizer, a learning rate of 0.001 and decay 0.0001.

**Models and parameterizations** We have an ordered response $Y \in \{1, \dots, K\}$ and covariates $X \in \mathbb{R}^p$. The DRIFT is parameterized with the following base distribution $F$ and conditional inverse flow $\phi^-$:

$$F(\cdot) \coloneqq \sigma(\cdot) = (1 + \exp(-\cdot))^{-1},$$

$$\phi^-(y, x) \coloneqq \phi_0^-(y) + \sum_{j=1}^p \rho_j(x_j),$$

$$\phi_0^-(y) \coloneqq (\mathbb{1}(y = 1), \dots, \mathbb{1}(y = K))^\top \boldsymbol{\theta}$$
$$\text{s.t. } \theta_1 < \theta_2 < \cdots < \theta_K \coloneqq +\infty,$$

where $\rho_j : [0, 1] \to \mathbb{R}, \quad j = 1, \dots, p$ are feed-forward neural networks as described above.

The POLR model is the same as above with the specialization that $\rho_j(x_j) \coloneqq x_j \beta_j, j = 1, \dots, p$ are linear functions.

### B.3.3 Survival Analysis

We used the London Fire Brigade data set analyzed in Taylor [2017]. The data describes the response times of the respective fire brigade in relation to spatiotemporal and economic features. We administratively censored excessive response times ($T > 1000$) and model the censored event time with a spatial effect (latitude and longitude), temporal effect (time of the day), and categorical features for the property type and the district name. The predictors $\rho_j$ are either linear effects for the categorical features or neural basis functions for the time and spatial features with structure as described in Section B.1. More precisely, the temporal effect is modeled with a univariate neural basis function with 64 and 12 units, each with ReLU activation function. The spatial effect is modeled by a bivariate NAM with 64, 32, 32, and 10 units each and ReLU activation function. Using these effects, we define the Cox proportional hazards model with smooth log cumulative baseline hazards in our DRIFT framework using the following parametrization:

$$F : z \mapsto 1 - \exp(-\exp(z))$$

$$\phi^-(y, x) \coloneqq \boldsymbol{a}(y)^\top \boldsymbol{\theta} + \sum_{j=1}^p \rho_j(x_j),$$

$$\text{s.t. } \theta_1 \leq \theta_2 \leq \dots \leq \theta_{M+1},$$

where $\Phi_{0,1}$ denotes the standard normal CDF and $\boldsymbol{a}$ a basis of polynomials in Bernstein form of order $M$. are feed-forward neural networks. We train the DRIFT using the Adam optimizer with a learning rate of 0.001 in a batch size of 32 for 250 epochs. As a comparison, we fit a piece-wise exponential additive model (PAM; Bender et al. [2018]) with equivalent feature effects using a thin-plate regression and tensor-product spline basis using `pammtools` Bender and Scheipl [2018]. The evaluation criterion is the reweighted integrated Brier Score introduced in Sonabend [2022] which is a proper scoring rule Gneiting and Raftery [2007]. Both learners are compared to an uninformative model, the Kaplan-Meier estimator (KM; Kaplan and Meier [1958]) as a baseline.

### B.3.4 Generalized Additive Models

We follow the data pre-processing of Rügamer [2023] and define both the NAM and the GAM using a Poisson outcome distribution. Both additive predictors use (neural-based) splines for date, population, temperature, and humidity. In addition, a tensor-product spline is used for latitude and longitude. Non-linear feature effects are defined by either

a) univariate basis functions as described in Section B.1 to resemble univariate splines,

b) two feed-forward neural networks as described in a) with 5 units in the last layer (instead of 1 unit) and then combined via a tensor-product followed by one last layer with 1 unit and no activation as well as no bias term.

The GAM uses thin-plate regression splines for univariate effects and a tensor-product spline version for the bivariate spatial effect. The DRIFT's inverse reference flow $\phi_0^-$ network is defined as a two-layer non-negative tanh-network with 10 neurons

each as well as positive weight constraint. All models are trained for a maximum of 250 epochs, batch size of 128, early stopping based on a 10% validation split with patience of 15 epochs, and Adam with a learning rate of 0.001.

### B.3.5 Time Series Regression

The electricity data set serves as a frequently chosen data set for state-of-the-art forecasting challenges [e.g. Wang et al., 2023, Wu et al., 2023]. It consists of records denoted in kilowatts at 15-minute intervals which we convert to kilowatt-hours. In DRIFT, the time lags enter linearly as location and scale effects on the standard Gaussian base distribution. The DRIFT's inverse reference flow $\phi_0^-$ network is defined as a three-layer non-negative tanh-network with 20 neurons for the first two layers, 5 neurons in the consecutive layer as well as positive weight constraint. The model for each univariate time series trains for a maximum of 10,000 epochs with a batch size of 256, and Adam as an optimizer with an initial learning rate of 0.0001. Early stopping is based on the validation set described in the main text with a patience of 10 epochs. For comparison, we employ the ARIMA model, a commonly used benchmark [e.g. Siami-Namini et al., 2018, Rügamer et al., 2023a]. We specify `auto.arima` such that the model complexity is found by a stepwise forward search based on the biased-corrected version of Akaike's Information Criterion (AICc), with initial lag values of $p = 12$ and $p = 24$ for the auto-regressive term, and $q = 0$ and $q = 3$ for the moving average term. We set the maximum number of model search steps to 25. The final model, which provides the log-scores obtained from the test set, is selected based on the ARIMA model with the lowest AICc on the validation set. DRIFT runs 5.4 hours. The auto.arima function in comparison has a runtime of 53 min.

### B.3.6 Benchmark Study

DR is defined by a parametric normal distribution with mean $\mu = \sum_{j=1}^p \rho_{\mu,j}$ and standard deviation $\sigma = \exp\left(\sum_{j=1}^p \rho_{\sigma,j}\right)$, with additive predictor structure as explained in the following paragraph. Training is done based on the negative log-likelihood. TMs fit into our DRIFT framework as a special case and are parameterized as follows:

$$F := \Phi_{0,1}$$

$$\phi^-(y, x) := \boldsymbol{a}(y)^\top \boldsymbol{\theta} + \sum_{j=1}^p \rho_j(x_j),$$

$$\text{s.t. } \theta_1 \leq \theta_2 \leq ... \leq \theta_{M+1},$$

where $\Phi_{0,1}$ denotes the standard normal CDF and $\boldsymbol{a}$ a basis of polynomials in Bernstein form of order $M$. DRIFT is defined by a location and scale effect outlined in the following paragraph. For $\phi_0^-$ see Section B.1. All models are trained using the Adam optimizer with a maximum of 1000 epochs, early stopping with a patience of 50.

**Predictor structure**

- **Structured** For comparisons of structured models, we use univariate thin-plate regression splines for DR and TMs for every feature and neural basis function splines as described in Section B.1.
- **Deep** For deep model comparisons, we model the predictors of DR, TM and DRIFT using four different multi-layer perceptron architectures:
    - Hidden(100,ReLU)-Dropout(0.1)-Hidden(1,Linear)
    - Hidden(100,ReLU)-Dropout(0.1)-Hidden(100,ReLU)-Dropout(0.1)-Hidden(1,Linear)
    - Hidden(20,ReLU)-Dropout(0.1)-Hidden(1,Linear)
    - Hidden(20,ReLU)-Dropout(0.1)-Hidden(20,ReLU)-Dropout(0.1)-Hidden(1,Linear)

    and for each method choose the best performing.
- **Semi-structured** For semi-structured comparisons, we use a combination of structured effects as outlined in the structured predictor section and one of the four deep neural networks as outlined in the deep predictor section.

### B.4 FURTHER RESULTS IN THE BENCHMARK STUDY

In case DRIFT and comparison methods define a deep or semi-structured model, we use a pre-defined set of four different deep architectures for all methods. The results for these different model specifications are given in Tables 4–2.

Table 4: Comparison results for different datasets (rows) and structured methods using neural basis functions (columns) showing the mean log-score (and standard deviation in brackets) based on a 10-fold cross-validation. The best methods per dataset are highlighted in bold.

| Dataset | DR | TM | DRIFT |
|---|---|---|---|
| Airfoil | -3.5 (1.1) | **-3.1** (0.1) | -3.5 (0.7) |
| Concrete | -3.3 (0.1) | -3.4 (0.3) | **-3.2** (0.3) |
| Diabetes | -8.5 (2.5) | **-5.5** (0.3) | **-5.5** (0.2) |
| Energy | -2.9 (0.3) | -2.8 (0.4) | **-2.3** (0.1) |
| Fish | **-1.3** (0.1) | -1.4 (0.2) | **-1.3** (0.3) |
| ForestF | -2.0 (0.3) | -1.5 (0.2) | **-1.4** (0.4) |
| Ltfsid | -7.7 (7.5) | -6.3 (0.4) | **-4.8** (0.2) |
| Naval | 4.1 (1.0) | 3.6 (0.3) | **5.1** (0.3) |
| Real | -1.4 (0.4) | **-0.8** (0.3) | -1.2 (1.1) |
| Wine | -0.2 (0.0) | 0.2 (0.2) | **4.2** (2.4) |
| Yacht | -1.1 (0.2) | -2.5 (3.0) | **-0.8** (0.1) |

Table 5: Comparison results for different datasets (rows) and *deep predictor* methods (columns) showing the mean log-score (and standard deviation in brackets) based on a 10-fold cross-validation. The best methods per dataset are highlighted in bold.

| Dataset | DR | TM | DRIFT |
|---|---|---|---|
| Airfoil | -4.4 (0.2) | **-3.1** (0.1) | -3.2 (0.3) |
| Concrete | -3.7 (0.1) | -3.4 (0.2) | **-3.2** (0.3) |
| Diabetes | -8.3 (1.2) | -5.7 (0.3) | **-5.2** (0.2) |
| Energy | -2.9 (0.1) | -3.1 (0.2) | **-2.5** (0.3) |
| Fish | **-1.3** (0.1) | -1.4 (0.2) | **-1.3** (0.3) |
| ForestF | -2.1 (0.5) | -1.6 (0.2) | **-1.2** (0.2) |
| Ltfsid | -5.7 (0.4) | -6.3 (0.4) | **-4.7** (0.1) |
| Naval | 4.7 (0.1) | 3.9 (0.2) | **5.0** (0.2) |
| Real | -1.5 (0.6) | -1.2 (0.5) | **-0.9** (0.7) |
| Wine | -0.2 (0.0) | 0.7 (0.2) | **4.2** (2.4) |
| Yacht | -1.6 (0.1) | -2.1 (1.6) | **-0.8** (0.1) |

## B.5   COMPUTATIONAL ENVIRONMENT

All real-world examples were conducted on conventional laptops and without GPU support.

