# OpenReview forum: "How Inverse Conditional Flows Can Serve as a Substitute for Distributional Regression"
_auai.org/UAI/2024/Conference — UAI 2024 poster_

### Official Review · Reviewer_YksT · 2024-03-20

**Q2-1 Originality-Novelty:** 3
**Q2-2 Correctness-Technical Quality:** 2
**Q2-5 Clarity Of Writing:** 3

**Q1 Summary And Contributions:**

This paper proposes a framework using inverse flow transformation for neural network representation of distributional regression models. This framework is offered to be applied where classical statistical models are suitable. For this purpose, a nonparametric transformation model is used, which utilizes monotone neural network. Finally, the performance of the framework is shown in different settings.

**Q2-3 Extent To Which Claims Are Supported By Evidence:**

3: Good: the main claims are supported by convincing evidence (in the form of adequate experimental evaluation, proofs, (pseudo-)code, references, assumptions).

**Q2-4 Reproducibility:**

2: Fair: key resources (e.g. proofs, code, data) are unavailable but key details (e.g. proof sketches, experimental setup) are sufficiently well-described for an expert to confidently reproduce the main results.

**Q3 Main Strengths:**

The paper is well written and easy to read and understand different sections. It contains an extensive experiment section where the proposed method is evaluated on different datasets. The figures are intuitive and explains the concepts clearly.

**Q4 Main Weakness:**

I am concerned about the technical contribution of this paper. The contributions are not specific. It is unclear what the new additions are compared to the normalizing flows model architecture.
Section 4 is the most important part in terms of new contributions. However, it seems that the authors discussed mainly the properties of the normalizing flow here. The difference between the proposed framework and NF should be highlighted.

**Q5 Detailed Comments To The Authors:**

* I would request the authors to share the technical contribution of their framework, which is not possible with an existing normalizing flow architecture.
* Can the authors provide some intuitions about why a neural network will be monotonic if the weights are strictly positive?
* What did the authors imply by model identifiability? Some details should be provided.
* Based on my understanding, the authors learn a network for each feature in Equation 3. In general we might have many features; how feasible is it to learn equal number of models as the features?
* Can we use MLP neural networks to learn the Gaussian mixture distribution in Figure 4?

**Q9 Complying With Reviewing Instructions:**

Yes

---

> ### Author Rebuttal · Authors · 2024-04-02
>
> > technical contribution [...] not specific. It is unclear what the new additions are compared to the normalizing flows model architecture. Section 4 is the most important part in terms of new contributions. However, it seems that the authors discussed mainly the properties of the normalizing flow here. The difference between the proposed framework and NF should be highlighted.
>
> DRIFTs are a subclass of normalizing flows. In Section 3, we outline how making such structural assumptions (i.e., making the class of flows smaller) yields gains in interpretability and oftentimes similar or better predictive performance (better than simpler statistical models, which assume, for instance, linearity, and better than more complex models, which may be harder to train). The goal of the proposed class is to interpolate between flexible flows and commonly less flexible distributional regression models from statistics. We outline our contributions in more detail in our reply to your comment below.
>
> > I would request the authors to share the technical contribution of their framework, which is not possible with an existing normalizing flow architecture.
>
> Our technical contributions are threefold:
> 1. We propose an interpretable location-scale family of inverse conditional flows and their implementation
> 2. Within the framework of neural network based transformation models, we are the first (to the best of our knowledge) to parameterize the reference flow in terms of invertible neural networks
> 3. Lastly, the connection between normalizing flows and distributional regression has not been made explicitly
>
> Our empirical contribution was to show that, for prediction, DRIFTs can be used instead of their statistical model counterpart and in many cases gain performance.
>
> > Can the authors provide some intuitions about why a neural network will be monotonic if the weights are strictly positive?
>
> Since the tanh function is monotonic and the positive weights preserve the order of their inputs, each layer in the network will maintain this monotonic relationship. That is, increasing the input to any neuron in the network will not decrease the output of that neuron, because the tanh function will monotonically increase (or stay constant), and the positive weights will not reverse the direction of this relationship.
>
> > What did the authors imply by model identifiability? Some details should be provided.
>
> Identifiability between the different predictors in (3) must be guaranteed to ensure that there is no overlap between the effects learned by the different functions $\rho_j(x_j)$. This is particularly crucial if interaction effects are included. For example for any $c\in\mathbb{R}\backslash 0$,
>
> $\rho_j(x_j) + \rho_k(x_k) = \rho_j(x_j) + c + \rho_k(x_k) - c = \tilde{\rho}_j(x_j) + \tilde{\rho}_k(x_k)$
>
> shows that two functions are generally not identifiable in their offset. To make them identifiable, we need to enforce sum-to-zero constraints for all functions $\rho_j$.
>
> > Based on my understanding, the authors learn a network for each feature in Equation 3. In general we might have many features; how feasible is it to learn equal number of models as the features?
>
> In neural networks, having more parameters than features is a very common phenomenon. The neural additive models $\rho_j$ learned in Eq. 3 are known to learn adaptive basis functions similar to the basis functions used in semi-parametric statistics. While the networks’ parameters are not identifiable, the functions $\rho_j$ themselves are (given certain constraints mentioned in our previous answer).
>
> > Can we use MLP neural networks to learn the Gaussian mixture distribution in Figure 4?
>
> We thank the reviewer for this interesting question. It is indeed possible to learn a Gaussian mixture model using an MLP. We, however, restricted our analysis in this case to models with an additive predictor to maintain interpretability.

---

### Official Review · Reviewer_jwGL · 2024-03-20

**Q2-1 Originality-Novelty:** 2
**Q2-2 Correctness-Technical Quality:** 3
**Q2-5 Clarity Of Writing:** 3

**Q1 Summary And Contributions:**

This paper investigates how the inverse conditional flow can unify many statistical distributional regression models. They proposed the DRIFT model, where the transformation model is replaced by the monotone neural networks. They also present a unified MLE framework to solve this DRIFT model and demonstrate its effectiveness among various statistical scenarios in the experiments.

**Q2-3 Extent To Which Claims Are Supported By Evidence:**

3: Good: the main claims are supported by convincing evidence (in the form of adequate experimental evaluation, proofs, (pseudo-)code, references, assumptions).

**Q2-4 Reproducibility:**

3: Good: key resources (e.g. proofs, code, data) are available and key details (e.g. proofs, experimental setup) are sufficiently well-described for competent researchers to confidently reproduce the main results.

**Q3 Main Strengths:**

1. Using inverse flow transformation to compare with distributional regression seems novel and well-motivated.

2. The experiments are extensive.

3. The paper is well-organized and easy to follow.

**Q4 Main Weakness:**

1. The technical contribution may be limited. The authors provided extensive discussion with related works in proposing their method, in which it seems their method is highly based on previous methods. Thus, it is hard to tell what the real technical contribution of this paper is.

2. Lack of theoretical analysis. Although I understand the proposed method tries to unify existing statistical models, it would always be better to have some theoretical results to interpret your proposed DRIFT model, instead of just unifying existing models.

**Q5 Detailed Comments To The Authors:**

Apart from the weakness I mentioned, I feel the title seems too large to make such a conclusion, i.e., inverse conditional flows can substitute for distributional regression. If the authors want to make this claim, it is necessary to have strong evidence to demonstrate what are the advantages of the proposed methods over the existing distributional regression. I understand the model capacity would be one, but  I think it is not enough.

**Q9 Complying With Reviewing Instructions:**

Yes

---

> ### Author Rebuttal · Authors · 2024-04-02
>
> > it seems their method is highly based on previous methods. Thus, it is hard to tell what the real technical contribution of this paper is.
>
> Our technical contributions are threefold:
> 1. We propose an interpretable location-scale family of inverse conditional flows and their implementation
> 2. Within the framework of neural network based transformation models, we are the first (to the best of our knowledge) to parameterize the reference flow in terms of invertible neural networks
> 3. Lastly, the connection between normalizing flows and distributional regression has not been made explicitly
>
> Our empirical contribution was to show that, for prediction and interpretability, DRIFTs can be used instead of their statistical model counterpart, and in many cases gain performance.
>
> > title seems too large to make such a conclusion [...] If the authors want to make this claim, it is necessary to have strong evidence to demonstrate what are the advantages of the proposed methods over the existing distributional regression.
>
> Thank you for this comment. We believe the title is phrased rather mildly by using “how” and “can”. This is demonstrated empirically by the proposed class of models (how), which does not perform worse (can) than their statistical counterparts -- and in some cases even better. We therefore believe that the wide range of applications and benchmarks presented in the paper supports this title, but we are open to other suggestions by the reviewer.

---

### Official Review · Reviewer_BpZk · 2024-03-25

**Q2-1 Originality-Novelty:** 3
**Q2-2 Correctness-Technical Quality:** 3
**Q2-5 Clarity Of Writing:** 3

**Q1 Summary And Contributions:**

In this paper, the authors propose a class of conditional flows called distributional regression using inverse flow transformations (DRIFT). This is a flexible class of models that closes the gap between normalizing flows and parametric transformation models for various outcome types. In more detail, by evaluating the likelihood comprised of the base CDF and parameterized inverse flow, DRIFT allows statistical models to be interpretable. Furthermore, the authors presented a range of numerical experiments to demonstrate whether DRIFT is a viable substitute for one or more established statistical approaches of similar complexity.

**Q2-3 Extent To Which Claims Are Supported By Evidence:**

2: Fair: the main claims are somewhat supported by evidence (but the experimental evaluation may be weak, or does not match entirely with the claims, important baselines may be missing, proofs contain important ideas but lack rigor, algorithmic details are only discussed superficially, references are imprecise, assumptions are not sufficiently motivated or explicated, etc.).

**Q2-4 Reproducibility:**

3: Good: key resources (e.g. proofs, code, data) are available and key details (e.g. proofs, experimental setup) are sufficiently well-described for competent researchers to confidently reproduce the main results.

**Q3 Main Strengths:**

I think the main strength of this work is that the authors propose a way to connect the advantages of normalizing flows and transformation models in a way that, to the best of my knowledge, has not appeared in the existing literature. Furthermore, they introduce well-designed assumptions and examples, and show that the proposed method works for various types of statistical data.

**Q4 Main Weakness:**

There are parts in the experiment that requires more detailed explanation to help the reader understand.
For example, in the results of section 5.1, the authors stated that an ordinal DRIFT outperforms the standard proportional odds logistic regression in log-score. However, it is necessary to clarify what "outperforming" means, whether it refers to a higher numerical value or more flexible (non-linear) relational modeling. Also, the difference in log-score does not seem large (-1.11 vs -1.12), and they need to provide additional explanation on this part.

Furthermore, in the conclusion section, they merely mentioned that the limitation can be solved even now ("~ can in principle be applied to DRIFT ~"), but it would be helpful if they could specify the reason why it couldn't be added to this paper.

**Q5 Detailed Comments To The Authors:**

Please check the issues highlighted in the "Main Weakness" section.

**Q9 Complying With Reviewing Instructions:**

Yes

---

> ### Author Rebuttal · Authors · 2024-04-02
>
> > parts in the experiment that requires more detailed explanation to help the reader understand. For example, in the results of section 5.1, the authors stated that an ordinal DRIFT outperforms the standard proportional odds logistic regression in log-score. However, it is necessary to clarify what "outperforming" means, whether it refers to a higher numerical value or more flexible (non-linear) relational modeling.
>
> Thank you for this comment. We fully agree and will rephrase the results saying that DRIFT and POLR perform on par.
>
> In general, our goal of the experiments was to provide empirical evidence that DRIFT can serve as a substitute for the statistical models in question. Hence being on par or outperforming another method does not change the conclusions of our work.
>
> > Furthermore, in the conclusion section, they merely mentioned that the limitation can be solved even now ("~ can in principle be applied to DRIFT ~"), but it would be helpful if they could specify the reason why it couldn't be added to this paper.
>
> These methods are applicable *in principle* and require assumptions and theoretical results beyond the ones we need for fitting DRIFTs and making predictions. In general, inference with neural network based models is a very difficult task and has been studied only for simpler, theoretically much more tractable architectures [Schmid-Hieber, 2017](https://arxiv.org/abs/1708.06633).

---

### Official Review · Reviewer_XFkE · 2024-03-25

**Q2-1 Originality-Novelty:** 2
**Q2-2 Correctness-Technical Quality:** 3
**Q2-5 Clarity Of Writing:** 3

**Q1 Summary And Contributions:**

While more classic statistical models (those focused on predicting the conditional mean) have been thoroughly studied through their neural network representations, those focused on predicting neural representations of distributional regression have not. The authors propose a new framework, called distributional regression using inverse flow transformations (DRIFT) to close this gap.
The idea behind normalized flows is to learn a feature-dependent transformation between the outcome and a latent variable with a fixed, simple distribution, such as the multivariate standard normal distribution.
Models in DRIFT empirically match the performance of several statistical methods in terms of estimation of partial effects, prediction, and aleatoric uncertainty quantification. Models in DRIFT have an additivity assumption on the conditional flow in terms of the features and neural basis functions to specify predictors for µ and σ. Several statistical models can be understood as DRIFT.

**Q2-3 Extent To Which Claims Are Supported By Evidence:**

2: Fair: the main claims are somewhat supported by evidence (but the experimental evaluation may be weak, or does not match entirely with the claims, important baselines may be missing, proofs contain important ideas but lack rigor, algorithmic details are only discussed superficially, references are imprecise, assumptions are not sufficiently motivated or explicated, etc.).

**Q2-4 Reproducibility:**

3: Good: key resources (e.g. proofs, code, data) are available and key details (e.g. proofs, experimental setup) are sufficiently well-described for competent researchers to confidently reproduce the main results.

**Q3 Main Strengths:**

The authors propose a new framework, called distributional regression using inverse flow transformations (DRIFT), that can be used as a neural network-based substitute for various distributional regression approaches (transformation, survival, and mixture models) in statistics. Models in DRIFT empirically match the performance of several statistical methods in terms of estimation of partial effects, prediction, and aleatoric uncertainty quantification.

**Q4 Main Weakness:**

It isn't clear how to how to reproduce the numerical experiments. Sharing the code and/or details on the implementation would be needed to improve the reproducibility of this work.

**Q5 Detailed Comments To The Authors:**

The proposed framework is interesting and empirically match the performance of several statistical methods in terms of estimation of partial effects, prediction, and aleatoric uncertainty quantification. However, it isn't clear how to how to reproduce the numerical experiments. Sharing the code and/or details on the implementation would be needed to improve the reproducibility of this work. It isn't clear also what does each experiment prove (perhaps less models but more thoroughly investigated may be better).

**Q9 Complying With Reviewing Instructions:**

Yes

---

> ### Author Rebuttal · Authors · 2024-04-02
>
> We thank the reviewer for the very positive review. We would like to point out that we have provided all the code as supplementary material (you will find it under *Supplementary Material* below the abstract on top of this page). We are also happy to open source all codes upon acceptance. We will also add a sentence to the main text mentioning our implementation and thank the reviewer for the comment.
>
> We hope that this addresses the reviewer’s concerns and if so, we would be very grateful if the reviewer could adjust the score accordingly.

---

### Meta-Review · Area_Chair_YKhA · 2024-04-18

This paper presents an interpretable location-scale family of inverse conditional flows for distributional regression. Basic idea is make the inverse conditional flows more flexible with a location adjusting function and a scale adjusting function in conditional flows. The proposed idea is demonstrate on several regression tasks.